# Altered mental status is a predictor of poor outcomes in COVID-19 patients: A cohort study

Abdallah S. Attia[1], Mohammad Hussein[1], Mohamed A. Aboueisha[1,2], Mahmoud Omar[1], Mohanad R. Youssef[1], Nicholas Mankowski[1], Michael Miller[1], Ruhul Munshi[1], Aubrey Swinford[1], Adam Kline[1], Therese Nguyen[1], Eman Toraih[1,3], Juan Duchesne[1], Emad Kandil[1] *

1 Department of Surgery, School of Medicine, Tulane University, New Orleans, Louisiana, United States of America, 2 Department of Otolaryngology-Head and Neck Surgery, Faculty of Medicine, Suez Canal University, Ismailia, Egypt, 3 Genetics Unit, Department of Histology and Cell Biology, Faculty of Medicine, Suez Canal University, Ismailia, Egypt

* ekandil@tulane.edu

**Data Availability Statement:** All relevant data are within the manuscript.

**Funding:** The author(s) received no specific funding for this work.

## Abstract

### Introduction

Several studies have described typical clinical manifestations, including fever, cough, diarrhea, and fatigue with COVID-19 infection. However, there are limited data on the association between the presence of neurological manifestations on hospital admission, disease severity, and outcomes. We sought to investigate this correlation to help understand the disease burden.

### Methods

We delivered a multi-center retrospective study of positive laboratory-confirmed COVID-19 patients. Clinical presentation, laboratory values, complications, and outcomes data were reported. Our findings of interest were Intensive Care Unit (ICU) admission, intubation, mechanical ventilation, and in-hospital mortality.

### Results

A total of 502 patients with a mean age of 60.83 ± 15.5 years, of them 71 patients (14.14%) presented with altered mental status, these patients showed higher odds of ICU admission (OR = 2.06, 95%CI = 1.18 to 3.59, $p = 0.01$), mechanical ventilation (OR = 3.28, 95%CI = 1.86 to 5.78, $p < 0.001$), prolonged (>4 days) mechanical ventilation (OR = 4.35, 95%CI = 1.89 to 10, $p = 0.001$), acute kidney injury (OR = 2.18, 95%CI = 1.28 to 3.74, $p = 0.004$), and mortality (HR = 2.82, 95%CI = 1.49 to 5.29, $p = 0.01$).

### Conclusion

This cohort study found that neurological presentations are associated with higher odds of adverse events. When examining patients with neurological manifestations, clinicians

**Competing interests:** The authors have declared that no competing interests exist.

should suspect COVID-19 to avoid delayed diagnosis or misdiagnosis and lose the chance to treat and prevent further transmission.

## Introduction

Since the Spanish flu pandemic in 1918, humankind hasn't encountered such an overwhelming health crisis created by the novel 2019 coronavirus disease (COVID-19) pandemic [1]. As of August 2, 2020, the World Health Organization (WHO) reports that 680,894 people have died around the world, spanning different countries, ethnicities, religions, and socioeconomic class. Furthermore, the Emergency Committee on COVID-19 unanimously admitted that the outbreak still poses a public health emergency of international concern [2].

COVID-19, caused by the severe acute respiratory syndrome coronavirus 2 (SARS-CoV-2), was initially thought to primarily infect the respiratory system with dyspnea, cough, expectoration, and chest pain being common presenting symptoms [3]. However, as the pandemic continues, it is has been shown that the virus affects a wide range of organ systems, including the gastrointestinal, hepatic, renal, and cardiovascular systems. Well-documented extrapulmonary findings in COVID-19 patients include diarrhea, nausea, vomiting, elevated liver enzymes, kidney dysfunction, elevated troponin and CK-MB, systolic dysfunction, and heart failure, especially those who developed a severe and critical illness [3–9].

Some articles have illustrated symptoms suggestive of potential nervous system involvement with many studies demonstrating anosmia, ageusia, dizziness, seizure, altered mental status (AMS), myalgia, headache, syncope, somnolence, and coma in COVID-19 patients [7–9, 10]. Neuropsychiatric manifestations, such as anxiety, depression, insomnia, and psychosis, are reported as well [7, 10]. Case reports and case series have reported para-infectious conditions including Guillain-Barre syndrome and ataxia [11]. However, there is limited data available describing neurological symptoms as presenting manifestations among COVID-19 patients. The purpose of this multi-center retrospective cohort study was to investigate whether presenting with neurological symptoms is a predictor of poor health outcomes and adverse events in COVID-19 positive patients.

## Methods

### Study design and population

This is a multi-center retrospective cohort study that was performed after acquiring Tulane University Institutional Review Board (IRB) approval. The patient data were collected on COVID-19 confirmed positive patients, who were admitted from March 20, 2020, to May 10, 2020, to Tulane Medical Center (TMC) and University Medical Center (UMC) in New Orleans, LA. The patient data was collected using Research Electronic Data Capture (REDCap) hosted at Tulane University Medical School. REDCap is a secure, web-based software platform designed to support data capture for research studies, providing an intuitive interface for validated data capture and audit trails for tracking data manipulation and export procedures [12]. Patients were divided into two groups: with and without altered mental status (AMS) which encompasses confusion, amnesia, loss of alertness, disorientation, defects in judgment or thought, unusual or strange behavior, poor regulation of emotions, and disruptions in perception, psychomotor skills, and behavior. Patients were diagnosed with AMS on admission by the attending physician and patients were not on any sedative agents at the time of diagnosis.

## Variables

Demographics, presenting symptoms, comorbidities, clinical notes, laboratory values, and health outcomes were extracted from the electronic medical records using a standardized data collection. Patient orientation and mental state were determined using Glasgow Coma Scale (GCS), patients with decreased GCS were considered to have Altered mental status. The severity of the disease was determined by two scoring systems: CURB-65 and Quick Sequential Organ Failure Assessment (qSOFA). The CURB-65 score is based on the presence of confusion, blood urea nitrogen level >19 mg/dL (>7 mmol/L), respiratory rate $\geq$30, blood pressure (systolic <90 mmHg or diastolic $\leq$60 mmHg), and age $\geq$65 years [13]. (2) The qSOFA score is based on a GCS <15, respiratory rate $\geq$22, and systolic blood pressure $\leq$100 [14].

## Outcomes

A comparison between patients with and without AMS was performed. Outcome measures investigated included disease course, Intensive Care Unit (ICU) admission, intubation, unplanned reintubation, mechanical ventilation, duration of mechanical ventilation, prolonged mechanical ventilation, ARDS, bacteremia, sepsis, acute kidney injury, length of hospital stay, and mortality.

## Statistical analysis

Data management was performed using SAS v9.4, while SPSS v26.0 was used for statistical analysis. Chi-square and Fisher's Exact tests were applied for categorical variables. Student's t and Mann-Whitney U tests were used for continuous variables. The two-sided $p$-value was set to be significant at <0.05. Multiple regression analysis was iterated using binary logistic regression models for all outcomes and cox hazard proportionate regression model for survival, adjusted by age, sex, obesity, and neuropsychiatric comorbidity.

# Results

## Demographics, comorbidities, and symptoms

We included a total of 502 COVID-19 confirmed positive patients with a mean age of 60.83 ± 15.5 years, and 238 patients (47.4%) were males. Their mean body mass index (BMI) was 33.32 ± 8.54 Kg/m$^2$, with 57.2% being obese having BMI >30 Kg/m$^2$ and 24.7% being overweight with a BMI >25 kg/m$^2$. Most participants were African Americans (74.7%). On admission, 37 (7.37%) participants were classified as asymptomatic, 71 patients (14.14%) presented with AMS including 25 patients without any other manifestations, and 394 (78.48%) presented with other non-specific, respiratory, and gastrointestinal symptoms without neuropsychiatric symptoms, **Fig 1**. Patients with AMS were significantly older (68.61 ± 16.36 years versus 59.56 ± 15.01 years, $p$ <0.001) and had lower BMI (30.91 ± 8.36 Kg/m$^2$ versus 33.83 ± 8.50 Kg/m$^2$, $p$ = 0.021). Patients with AMS were less likely to be African American (66.2% versus 76.1%, $p$ = 0.023), obese (45.1% versus 59.2%, $p$ = 0.028), and have shortness of breath on admission (33.8% versus 58.2%, $p$ < 0.001), **Table 1**.

## Clinical assessment

Patients with AMS had a higher qSOFA score (1.40 ± 0.76 versus 0.58 ± 0.61, $p$ <0.001), CURB-65 score (2.63 ± 1.07 versus 1.22 ± 1.00, $p$ <0.001) and lower GCS (9.41 ± 4.0815.00 ± 0.00, $p$<0.001) compared to non-AMS cohorts, **Table 1**.

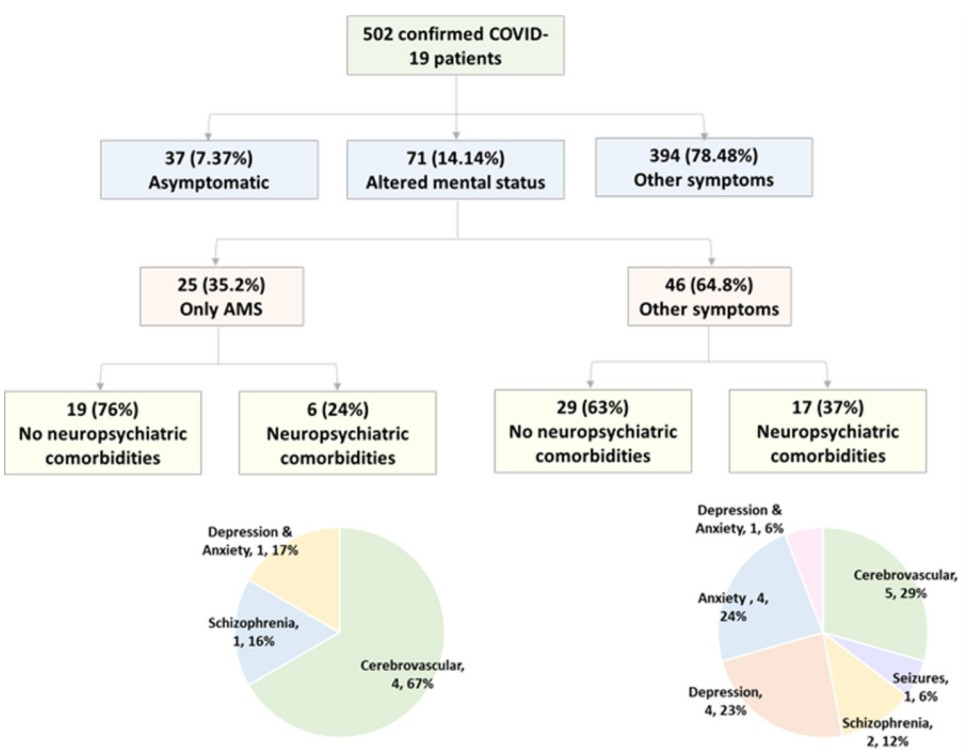

**Fig 1. Hierarchical classification of patients according to neuropsychiatric symptoms and comorbidities.**

## Laboratory findings

Patients presented with AMS had a lower $PaO_2/FiO_2$ (192.89 ± 114.26 versus 265.26 ± 103.25, $p = 0.003$) and higher white blood cell count (9.69 ± 6.03 versus 7.85 ± 5.26, $p = 0.013$), neutrophil-to-lymphocyte ratio (10.78 ± 9.38 versus 7.18 ± 9.76, $p = 0.007$), blood urea nitrogen (31.15 ± 19.23 versus 25.01 ± 20.39, $p = 0.02$), and C-reactive protein (CRP) (93.61 ± 81.27 versus 32.36 ± 47.69, $p = 0.001$) compared to patients without AMS, **Table 1**.

## Adverse events

Patients presented with AMS showed worse outcomes compared to patients without AMS. Presenting with AMS was associated with higher rates of ICU admission (42.3% versus 27.7%, $p = 0.017$), intubation (45.1% versus 27%, $p = 0.003$), mechanical ventilation (47.9% versus 24.6%, $p < 0.001$), unplanned reintubation (35.3% versus 9.5%, $p = 0.012$), sepsis (23.9% versus 13.5%, $p = 0.03$), and acute kidney injury (N = 26, 36.6% versus N = 90, 20.9%, $p = 0.006$), **Table 2**.

## Mortality and length of stay

Patients presented with AMS had a higher rate of mortality (29.6% versus 11.6%, $p < 0.001$), earlier death (9.05 ± 5.08 days versus 15.07 ± 9.27, $p = 0.007$), and longer duration on ventilators (4.63 ± 6.08 days versus 1.43 ± 3.51 days, $p < 0.001$), **Table 2**.

## Predictors risk factor for poor outcomes in patients with AMS

The multiple regression analysis was adjusted for age, sex, obesity, and neuropsychiatric comorbidities. This analysis revealed that patients presented with AMS had higher odds of

**Table 1. Characteristics of COVID-19 patients at admission.**

| Characteristics | | Non-AMS (n = 431) | AMS (n = 71) | P value |
|---|---|---|---|---|
| **Demographic data** | | | | |
| Age | Mean ± SD | 59.56 ± 15.01 | 68.61 ± 16.36 | **<0.001** |
| | 18–49 years | 97 (22.7) | 12 (17.1) | **<0.001** |
| | 50–64 years | 172 (40.2) | 12 (17.1) | |
| | ≥ 65 years | 159 (37.1) | 46 (65.7) | |
| Sex | Female | 228 (53.1) | 33 (47.1) | 0.36 |
| | Male | 201 (46.9) | 37 (52.9) | |
| Race | African American | 328 (76.1) | 47 (66.2) | **0.023** |
| | White | 72 (16.7) | 12 (16.9) | |
| | Not Reported | 31 (7.2) | 12 (16.9) | |
| BMI, kg/m$^2$ | Mean ± SD | 33.83 ± 8.50 | 30.91 ± 8.36 | **0.021** |
| Smoking | None | 302 (70.1) | 44 (62) | 0.26 |
| | Past smoker | 90 (20.9) | 21 (29.6) | |
| | Current smoker | 39 (9) | 6 (8.5) | |
| **Chief complaints** | | | | |
| Asymptomatic | Asymptomatic | 37 (8.6%) | NA | NA |
| Non-specific | Fever | 97 (22.5) | 11 (15.5) | 0.21 |
| | Fatigue/weakness | 31 (7.2) | 1 (1.4) | 0.06 |
| | Myalgia/FLS | 28 (6.5) | 5 (7) | 0.79 |
| | Headache | 6 (1.4) | 1 (1.4) | 0.99 |
| Respiratory symptoms | Shortness of breath | 251 (58.2) | 24 (33.8) | **<0.001** |
| | Cough | 104 (24.1) | 11 (15.5) | 0.12 |
| | Chest pain | 13 (3) | 0 (0) | 0.23 |
| GIT symptoms | Nausea, vomiting, diarrhea | 28 (6.5) | 3 (4.2) | 0.60 |
| **Comorbidities** | | | | |
| Neuropsychiatric | Overall | 99 (23) | 23 (32.4) | 0.10 |
| | Cerebrovascular disease | 29 (6.7) | 9 (12.7) | 0.09 |
| | Seizures | 6 (1.4) | 1 (1.4) | 0.99 |
| | Mood disorders | 43 (10) | 7 (9.9) | 0.97 |
| | Anxiety disorder | 24 (5.6) | 6 (8.5) | 0.41 |
| | Schizophrenia | 10 (2.3) | 3 (4.2) | 0.40 |
| Other comorbidities | Overall | 378 (87.7) | 64 (90.1) | 0.69 |
| | Obesity | 255 (59.2) | 32 (45.1) | **0.028** |
| | Hypertension | 303 (70.3) | 54 (76.1) | 0.39 |
| | Diabetes | 185 (42.9) | 26 (36.6) | 0.36 |
| | Chronic heart failure | 39 (9) | 11 (15.5) | 0.13 |
| | Arrhythmia | 40 (9.3) | 8 (11.3) | 0.66 |
| | Coronary artery disease | 39 (9) | 10 (14.1) | 0.20 |
| | Asthma | 66 (15.3) | 6 (8.5) | 0.15 |
| | COPD | 30 (7) | 6 (8.5) | 0.62 |
| | Chronic kidney disease | 63 (14.6) | 13 (18.3) | 0.47 |
| | Cancer | 46 (10.7) | 9 (12.7) | 0.68 |
| **Clinical assessment** | | | | |
| Severity | qSOFA score | 0.58 ± 0.61 | 1.40 ± 0.76 | **<0.001** |
| | CURB65 score | 1.22 ± 1.00 | 2.63 ± 1.07 | **<0.001** |
| Orientation | Glasgow coma score | 15.00 ± 0.00 | 9.41 ± 4.08 | **<0.001** |

*(Continued)*

**Table 1.** (Continued)

| Characteristics | | Non-AMS (n = 431) | AMS (n = 71) | P value |
|---|---|---|---|---|
| Vital signs | Temperature (F) | 99.65 ± 1.72 | 98.75 ± 1.67 | **<0.001** |
| | Pulse rate | 91.21 ± 19.06 | 83.18 ± 21.20 | **0.002** |
| | Systolic blood pressure | 126.34 ± 20.18 | 122.84 ± 24.56 | 0.21 |
| | Diastolic blood pressure | 74.33 ± 14.82 | 70.07 ± 15.77 | **0.033** |
| | Mean arterial pressure | 101.10 ± 18.08 | 100.54 ± 21.13 | 0.82 |
| | Respiratory rate | 22.25 ± 7.39 | 21.39 ± 6.16 | 0.37 |
| ABG findings | $SaO_2$ | 92.87 ± 7.88 | 94.26 ± 7.37 | 0.18 |
| | pH respiratory | 7.26 ± 1.02 | 7.40 ± 0.07 | 0.45 |
| | $PaCO_2$ | 39.27 ± 13.99 | 38.17 ± 9.62 | 0.67 |
| | $PaO_2$ | 86.44 ± 60.70 | 99.33 ± 99.46 | 0.34 |
| | Anion gap | 11.73 ± 10.53 | 12.90 ± 3.55 | 0.56 |
| | Lactic acid | 54.82 ± 108.90 | 33.87 ± 85.16 | 0.66 |
| | $HCO_3$ | 24.96 ± 3.19 | 22.91 ± 4.94 | **0.017** |
| | $FiO_2$ (%) | 33.59 ± 24.87 | 58.57 ± 33.08 | **<0.001** |
| | $PaO_2/FiO_2$ ratio | 265.26 ± 103.25 | 192.89 ± 114.26 | **0.003** |
| **Laboratory findings** | | | | |
| Complete blood picture | White blood cells (x$10^9$/L) | 7.85 ± 5.26 | 9.69 ± 6.03 | **0.013** |
| | Hemoglobin (g/dl) | 12.11 ± 2.08 | 11.98 ± 2.07 | 0.66 |
| | Hematocrit (%) | 36.32 ± 5.91 | 35.95 ± 5.95 | 0.69 |
| | Platelet count (x$10^9$/L) | 237.75 ± 100.14 | 239.22 ± 126.82 | 0.92 |
| | Neutrophil count (x$10^9$/L) | 6.58 ± 8.78 | 8.85 ± 11.16 | 0.07 |
| | Lymphocyte count (x109/L) | 1.34 ± 1.94 | 1.08 ± 0.77 | 0.29 |
| | Neutrophil lymphocyte ratio | 7.18 ± 9.76 | 10.78 ± 9.38 | **0.007** |
| Electrolytes | Serum sodium (mmol/L) | 209.29 ± 937.39 | 138.85 ± 5.20 | 0.59 |
| | Serum potassium (mmol/L) | 4.09 ± 1.14 | 4.06 ± 0.65 | 0.84 |
| | Serum chloride (mmol/L) | 101.46 ± 5.24 | 101.75 ± 14.86 | 0.82 |
| | Calcium corrected (mmol/L) | 9.02 ± 0.67 | 8.90 ± 0.80 | 0.18 |
| Glycemic profile | Random blood sugar (mg/dl) | 144.47 ± 84.51 | 158.75 ± 97.36 | 0.22 |
| | HbA1c (%) | 7.94 ± 3.05 | 5.70 ± 0.00 | 0.49 |
| Renal function test | Blood urea nitrogen (mg/dl) | 25.01 ± 20.39 | 31.15 ± 19.23 | **0.025** |
| | Serum creatinine (mg/dl) | 1.74 ± 2.00 | 2.03 ± 2.19 | 0.28 |
| Liver function test | Total protein (g/dl) | 6.98 ± 0.74 | 6.88 ± 0.55 | 0.48 |
| | Albumin (g/dl) | 3.29 ± 0.55 | 3.06 ± 0.67 | **0.017** |
| | Bilirubin (mg/dl) | 0.61 ± 0.45 | 0.69 ± 0.49 | 0.35 |
| | Alkaline phosphatase (U/L) | 75.41 ± 44.18 | 74.27 ± 33.48 | 0.90 |
| | AST (U/L) | 48.16 ± 33.70 | 54.27 ± 43.07 | 0.41 |
| | ALT (U/L) | 35.52 ± 29.73 | 33.43 ± 26.72 | 0.73 |
| Cardiac marker | Troponin (ng/ml) | 3.63 ± 16.61 | 0.78 ± 1.55 | 0.62 |
| Inflammatory markers | C-reactive protein (mg/dl) | 32.36 ± 47.69 | 93.61 ± 81.27 | **0.001** |
| | Procalcitonin (ng/ml) | 11.64 ± 66.58 | 0.30 ± 0.30 | 0.50 |
| | Ferritin (ng/ml) | 989.76 ± 1,922.35 | 1,428.71 ± 2,825.35 | 0.38 |

AMS: altered mental status, FLS: Flu-like symptoms, GIT: gastrointestinal tract, NA: not applicable.

Data are presented as mean and standard deviation or frequency and percentage. BMI: body mass index. SaO2: oxygen saturation, PaO2: partial pressure of oxygen, PaCO2: partial pressure of carbon dioxide, HCO3: bicarbonate, FiO2: Fraction of inspired oxygen, AST: Aspartate transaminase, ALT: alanine transaminase, HbA1c: glycosylated hemoglobin. Chi-square, Fisher's Exact, Student's t, or Mann-Whitney U tests were used. $P$-value at $<0.05$ was considered significant.

**Table 2. Outcomes of COVID-19 patients with and without altered mental status.**

| Characteristics | | Non-AMS (n = 431) | AMS (n = 71) | P-value |
|---|---|---|---|---|
| | | N (%) or M±SD | N (%) or M±SD | |
| Hospital admission | Floor | 312(72.3) | 41 (57.7) | **0.017** |
| | ICU | 119 (27.7) | 30 (42.3) | |
| Procedures | Mechanical ventilation | 106 (24.6) | 34 (47.9) | **<0.001** |
| | Require intubation | 115 (27) | 32 (45.1) | **0.003** |
| | Extubation* | 84 (73) | 17 (53.1) | **0.032** |
| Develop complications | Negative | 215 (49.9) | 31 (43.7) | 0.37 |
| | Positive | 216 (50.1) | 40 (56.3) | |
| Type of complications | ARDS | 139 (32.3) | 21 (29.6) | 0.68 |
| | Unplanned reintubation** | 8 (9.5) | 6 (35.3) | **0.012** |
| | Sepsis | 58 (13.5) | 17 (23.9) | **0.030** |
| | Bacteremia | 32 (7.4) | 2 (2.8) | 0.20 |
| | Acute kidney injury | 90 (20.9) | 26 (36.6) | **0.006** |
| Mortality | Alive | 381 (88.4) | 50 (70.4) | **<0.001** |
| | Dead | 50 (11.6) | 21 (29.6) | |
| Death location*** | Floor | 3 (7) | 3 (14.3) | 0.38 |
| | ICU | 40 (93) | 18 (85.7) | |
| Days to event | Renal failure | 2.44 ± 3.19 | 2.67 ± 4.62 | 2.44 |
| | ARDS | 1.97 ± 2.10 | 0.60 ± 1.34 | 1.97 |
| | Sepsis | 1.19 ± 2.14 | 0.75 ± 1.50 | 1.19 |
| | Extubation | 8.93 ± 5.47 | 8.67 ± 7.30 | 0.90 |
| | Death | 15.07 ± 9.27 | 9.05 ± 5.08 | **0.007** |
| Ventilation days | Overall | 1.43 ± 3.51 | 4.63 ± 6.08 | **<0.001** |
| | Discharged | 0.90 ± 2.78 | 2.82 ± 6.88 | **0.017** |
| | Deceased | 6.33 ± 5.38 | 6.83 ± 4.12 | 0.76 |
| Total LOS | Overall | 12.24 ± 11.11 | 10.89 ± 9.79 | 0.40 |
| | Discharged | 11.89 ± 11.29 | 12.09 ± 11.84 | 0.92 |
| | Deceased | 14.81 ± 9.45 | 9.05 ± 5.08 | **0.011** |
| ICU LOS | Overall | 9.38 ± 7.60 | 8.46 ± 5.97 | 0.56 |
| | Discharged | 7.96 ± 6.52 | 10.29 ± 9.01 | 0.39 |
| | Deceased | 12.27 ± 8.84 | 7.86 ± 4.70 | **0.040** |

Data are presented as mean and standard deviation (M±SD) or frequency and percentage between parentheses.

*Percentage among intubated patients

**Percentage among extubated patients

*** data for the death location for 7 patients were missing.

ICU admission (OR = 2.06, 95% CI = 1.18 to 3.59, $p = 0.010$), intubation (OR = 2.53, 95% CI = 1.44 to 4.43, $p = 0.001$), mechanical ventilation (OR = 3.28, 95% CI = 1.86 to 5.78, $p < 0.001$), prolonged (>4 days) ventilation (OR = 4.35, 95% CI = 1.86 to 10, $p = 0.001$), sepsis (OR = 2.02, 95% CI = 1.10 to 3.73, $p = 0.024$), acute kidney injury (OR = 2.18, 95% CI = 1.28 to 3.74, $p = 0.004$), and mortality (HR = 2.81, 95% CI = 1.49 to 5.29, $p = 0.001$), **Fig 2**.

## Discussion

The COVID-19 pandemic has been a serious health emergency and has caused an unprecedented international disaster while creating damaging social, economic, and political consequences that will likely have devastating long-term effects. Following disease-control

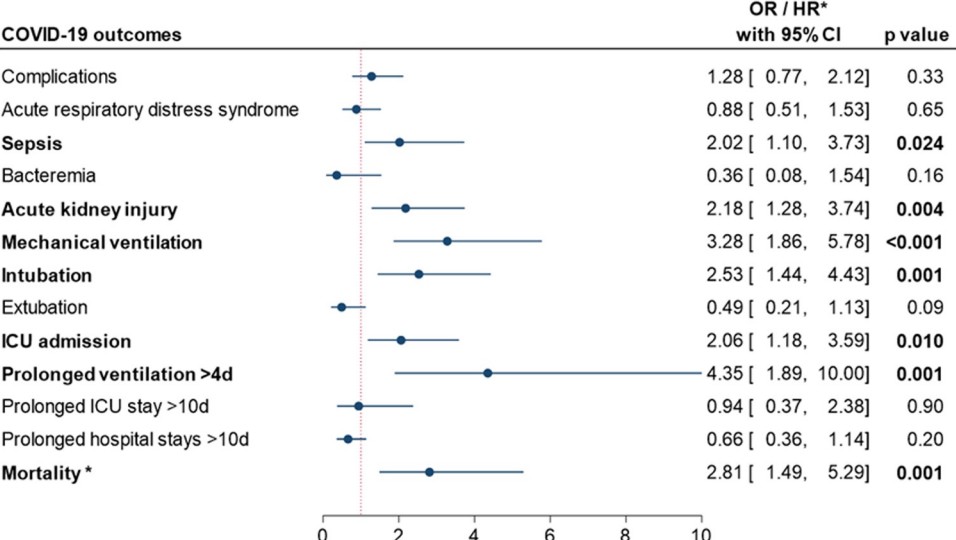

| COVID-19 outcomes | | OR / HR* with 95% CI | p value |
|---|---|---|---|
| Complications | | 1.28 [ 0.77, 2.12] | 0.33 |
| Acute respiratory distress syndrome | | 0.88 [ 0.51, 1.53] | 0.65 |
| **Sepsis** | | 2.02 [ 1.10, 3.73] | **0.024** |
| Bacteremia | | 0.36 [ 0.08, 1.54] | 0.16 |
| **Acute kidney injury** | | 2.18 [ 1.28, 3.74] | **0.004** |
| **Mechanical ventilation** | | 3.28 [ 1.86, 5.78] | **<0.001** |
| **Intubation** | | 2.53 [ 1.44, 4.43] | **0.001** |
| Extubation | | 0.49 [ 0.21, 1.13] | 0.09 |
| **ICU admission** | | 2.06 [ 1.18, 3.59] | **0.010** |
| **Prolonged ventilation >4d** | | 4.35 [ 1.89, 10.00] | **0.001** |
| Prolonged ICU stay >10d | | 0.94 [ 0.37, 2.38] | 0.90 |
| Prolonged hospital stays >10d | | 0.66 [ 0.36, 1.14] | 0.20 |
| **Mortality *** | | 2.81 [ 1.49, 5.29] | **0.001** |

**Fig 2. Impact of altered mental status as a predictor risk factor for poor outcomes.** Multiple regression analysis was iterated using binary logistic regression models for all outcomes and cox hazard proportionate regression model for survival, adjusted by age, sex, obesity, and neuropsychiatric comorbidity. Results are reported as odds ratio (OR) for all outcomes or hazard ratio (HR*) for survival.

guidelines, identifying risk factors, and recognizing different manifestations of COVID-19 infection is critical to deterring the spread and progression to severe disease. In response to this crisis, we conducted a retrospective cohort study on 502 hospitalized laboratory-confirmed COVID-19 patients to identify the outcomes associated with neurological symptoms, specifically AMS.

Angiotensin-converting enzyme 2 (ACE2) is the host functional receptor recognized by viral protein (spike) and allows the SARS-CoV-2 to enter the cell [15]. It is documented that SARS-CoV-2 has a higher affinity for ACE2 compared to its predecessor, SARS-CoV, explaining the higher rates of transmission. Due to the high presence of ACE2 on type II alveolar epithelial cells, the lung is the primary target and most vulnerable organ. However, the expression of ACE2 is ubiquitous, presenting in other multiple human tissues, including adipose tissue and nervous system [7, 16–18]. Due to increased expression in ACE2 in adipose tissue, obese individuals, could develop an explosive systemic inflammatory response, possibly contributing to the development of a more severe form of the disease [19]. Expression of ACE2 on glial cells, neurons, and capillary endothelial cells suggests that SARS-CoV-2 may invade the central nervous system (CNS) via direct invasion or cerebrovascular endothelium [7, 20–22].

Paniz-Mondolfi et al. reported the presence of SARS-CoV-2 in brain tissue from the post-mortem examination of a COVID-19 patient by implementing a transmission electron microscope. The viral particles were detected in the frontal lobe and matched the structural characteristics of SARS-CoV-2. Notably, these viral particles were found in the small vesicles of endothelial cells, which supports CNS invasion via hematogenous pathways may be a cause of the rapid progression of neurological symptoms [23]. Additionally, SARS-CoV-2 was identified in the cerebrospinal fluid (CSF) via polymerase chain reaction (PCR) in a male patient suffering from impaired consciousness and transient generalized seizures, with typical meningitis and encephalitis characteristics shown on the magnetic resonance imaging (MRI). Interestingly, this case presented with negative PCR results in the nasopharyngeal swab, which indicated that CNS invasion might have occurred in the early phase of COVID-19 infection [24]. It is also suggested

that direct invasion into the neuronal cells and retrograde transport from the olfactory bulb may be the pathophysiology of anosmia experienced by some COVID-19 patients [22, 25].

To our knowledge, this is the first cohort study comparing outcomes between COVID-19 patients with and without neurologic symptoms, specifically AMS, while utilizing validated scoring systems (GCS, CURB-65, and qSOFA). Our initial univariate analysis showed that 14.14% of the COVID-19 patients presented to the hospital with AMS. These patients had higher rates of developing adverse events such as ICU admission, intubation, mechanical ventilation, prolonged ventilation, extended hospital stay, and mortality. But they had a lower rate of shortness of breath which could be explained due to the decreased reporting of symptoms with patients with AMS [26]. Out of the patients presenting with AMS in our cohort, 25 (35.2%) presented with no other symptoms. Mao et al. also reported that neurological symptoms, including impaired consciousness, occurred early in the disease course, sometimes preceding typical respiratory symptoms [7]. This suggests that neurological symptoms, such as AMS, may be signs of impending clinical decline in the early stages of COVID-19.

The patients with AMS presented with comparatively more severe disease, shown by the significantly higher CURB-65 and qSOFA scores at the time of admission. Additionally, patients with AMS presented with significantly higher neutrophil-to-lymphocyte ratio and CRP, which are risk factors of poor health outcomes and possible predictors of severe disease [27–30]. However, it cannot be definitively determined if AMS is a result of a more severe disease or that neurological involvement, manifesting as AMS, is causing more severe features of COVID-19.

Older age, obesity, and being African American are all associated with poor health outcomes in COVID-19 patients [19, 31–34]. The prevalence of obese patients in our study was 57% which is higher than Louisiana's average (35%) [35]. It should be acknowledged that the AMS cohort has a significantly older average age. However, when adjusted for age, AMS remained more associated with poor health outcomes, **Fig 2**. Notably, patients with AMS were less likely to be African American or have obesity, which strengthens the argument that AMS may be an independent risk factor for poor health outcomes in COVID-19 patients. But we were unable to explain the reason why.

There are many established etiologies of AMS, including neurologic, toxicologic, trauma, psychiatric, and infectious [34]. Neuropsychiatric comorbidities could predispose patients to develop AMS. In our cohort, there was no significant difference in the prevalence of neuropsychiatric comorbidities between patients with and without AMS. Additionally, AMS was remained associated with poor health outcomes when the analysis was adjusted for neuropsychiatric comorbidities, **Fig 2**. Dehydration is also associated with developing AMS, especially in elderly patients [36]. Diarrhea and vomiting are possible causes of dehydration, and there was no significant difference in the prevalence of these symptoms between the study groups. Additionally, AMS secondary to hospital-induced delirium can be ruled out, since all these patients presented with AMS at admission.

Further studies are needed to determine if presenting with only AMS is also associated with poor health outcomes, and why they are presented less in obese and African American patients, which may further strengthen the argument that it should be considered an independent risk factor for poor health outcomes. Limiting data analysis to data upon admission is a potential limitation, which makes it prone to missing data such as CT-scans, MRI, and CSF analysis, which are vital in identifying brain injury and signs of neurological invasion of SAR-CoV-2.

## Author Contributions

**Conceptualization:** Abdallah S. Attia, Mohammad Hussein, Mohanad R. Youssef, Emad Kandil.

**Data curation:** Abdallah S. Attia, Mohammad Hussein, Mahmoud Omar, Mohanad R. Youssef, Aubrey Swinford, Adam Kline, Therese Nguyen, Emad Kandil.

**Formal analysis:** Abdallah S. Attia, Mohammad Hussein, Eman Toraih.

**Funding acquisition:** Abdallah S. Attia.

**Investigation:** Abdallah S. Attia, Mohanad R. Youssef, Eman Toraih.

**Methodology:** Abdallah S. Attia, Mohammad Hussein, Eman Toraih, Emad Kandil.

**Project administration:** Abdallah S. Attia.

**Resources:** Abdallah S. Attia, Eman Toraih, Emad Kandil.

**Software:** Eman Toraih.

**Supervision:** Eman Toraih, Juan Duchesne, Emad Kandil.

**Validation:** Eman Toraih.

**Visualization:** Abdallah S. Attia, Mohamed A. Aboueisha.

**Writing – original draft:** Abdallah S. Attia, Mohamed A. Aboueisha, Mahmoud Omar, Nicholas Mankowski.

**Writing – review & editing:** Abdallah S. Attia, Mohamed A. Aboueisha, Mahmoud Omar, Nicholas Mankowski, Michael Miller, Ruhul Munshi, Aubrey Swinford, Adam Kline, Therese Nguyen, Eman Toraih, Juan Duchesne, Emad Kandil.

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
