## [Decision Letter · Decision Letter 0]

10 Mar 2021

PONE-D-21-02993

Altered mental status is a predictor of poor outcomes in COVID-19 patients: A cohort study

PLOS ONE

Dear Dr. Aboueisha,

Thank you for submitting your manuscript to PLOS ONE. After careful consideration, we feel that it has merit but does not fully meet PLOS ONE’s publication criteria as it currently stands. Therefore, we invite you to submit a revised version of the manuscript that addresses the points raised during the review process.

We look forward to receiving your revised manuscript.

Kind regards,

Tai-Heng Chen, M.D.

Academic Editor

PLOS ONE

2. Please include a separate caption for each figure in your manuscript.

3. Please include your tables as part of your main manuscript and remove the individual files. Please note that supplementary tables should be uploaded as separate "supporting information" files.

Reviewers' comments:

Reviewer's Responses to Questions

**Comments to the Author**

1. Is the manuscript technically sound, and do the data support the conclusions?

Reviewer #1: Partly

Reviewer #2: Yes

2. Has the statistical analysis been performed appropriately and rigorously? 

Reviewer #1: Yes

Reviewer #2: Yes

3. Have the authors made all data underlying the findings in their manuscript fully available?

Reviewer #1: Yes

Reviewer #2: Yes

4. Is the manuscript presented in an intelligible fashion and written in standard English?

Reviewer #1: No

Reviewer #2: Yes

5. Review Comments to the Author

Reviewer #1: In this study, authors investigated the association between altered mental status and outcomes in COVID- 19 patients. They found that patients with altered mental status had poor outcomes compared to those patients without altered mental status. Despite the results, I think that there are some concerns.

1. The altered mental status is a vague term and definition. Please clarify the definition of altered mental status.

2. Depth of sedation could be related to altered mental status. Please provide the information of sedative agents.

3. Please provide the causes of altered mental status such as seizure, sepsis associated encephalopathy, metabolic encephalopathy and stroke.

4. When did evaluate the altered mental status during hospitalization? Please provide this information.

Reviewer #2: My comments are below:

Comment 1: In the line 81 and 82, 'Patients were divided into two groups: with and without altered mental status (AMS).' The definition of AMS was not clear. In order to get a more objective results, an operational definition for AMS is essential.

Comment 2:In the line 109 and 110, '...(BMI) was 33.32...with 57.2% being obese and 24.7 being overweight.' It's better to more clearly define 'obese' and 'overweight' . In addition, obesity is a critical issue for COVID-19 patients, it's also suitable to provide the BMI of normal population in New Orleans for comparison if possible.

Comment 3: The data of table 2 needs to be carefully corrected. For example, the meaning of the data in every parentheses made me confusing (e.g. in the first row of ED disposition : 134 (77) , 31 (72.1); however in the first row of Hospital admission: 310 (72.3), 41(57.7) ; the calculation method of each parentheses was inconsistent ! the inconsistency also noted in the sub-table of Procedures . Besides, the case numbers of hospital admission in Non-AMS group was 310+119=429 , but in the first row : Non-AMS (n=431), the authors should explain the discrepancy).

Comment 4:　In line 211-212 '... patients with AMS were less likely to be African American or have obesity...', but the authors didn't provide possible reasons for the differences. Besides, I wonder the body surface area might also differed between non-AMS and AMS groups.

Comment 5: According to fig 2, the AMS group was more likely to be intubated and needs mechanical ventilation, but in table 1, the respiratory symptoms of AMS group seemed to be less, especially ' Shortness of breath', this contradiction needs to be discussed.

6. PLOS authors have the option to publish the peer review history of their article (what does this mean?). If published, this will include your full peer review and any attached files.

Reviewer #1: No

Reviewer #2: **Yes: **Chi-Hsiang Chou

---

## [Author Response · Author response to Decision Letter 0]

14 Apr 2021

Comment 1: In the line 81 and 82, 'Patients were divided into two groups: with and without altered mental status (AMS).' The definition of AMS was not clear. In order to get a more objective results, an operational definition for AMS is essential.

Author Response Thank you for the comment. The Definition has been clarified in the manuscript and it is as following: altered mental status (AMS) which encompasses confusion, amnesia, loss of alertness, disorientation, defects in judgment or thought, unusual or strange behavior, poor regulation of emotions, and disruptions in perception, psychomotor skills, and behavior. Line (82-85)

Comment 2: In the line 109 and 110, '...(BMI) was 33.32...with 57.2% being obese and 24.7 being overweight.' It's better to more clearly define 'obese' and 'overweight' . In addition, obesity is a critical issue for COVID-19 patients, it's also suitable to provide the BMI of normal population in New Orleans for comparison if possible.

Author Response Thank you for pointing this out. We agree with this comment. Therefore, we have added the definition of the Obesity and overweight in the results. (line114,115)

The % of Obese population in Louisiana has been added to the discussion where our cohort had higher percentage of obese patient than the average of Louisiana population. Changes has been marked in the manuscript. (215-217)

Comment 3(A): The data of table 2 needs to be carefully corrected. For example, the meaning of the data in every parentheses made me confusing (e.g. in the first row of ED disposition: 134 (77) , 31 (72.1); however in the first row of Hospital admission: 310 (72.3), 41(57.7) ; the calculation method of each parentheses was inconsistent ! the inconsistency also noted in the sub-table of Procedures . 

Author Response Thank you for pointing this out. The ED depositions data account for patients how came through the Emergency Room a total of 166 in non AMS and 43 for AMS and between the parentheses represent the (%) of the ER cases but these data don’t show any significance, so it was removed to avoid any confusion. (table 2) 

Comment 3(B) the calculation method of each parentheses was inconsistent! the inconsistency also noted in the sub-table of Procedures

Author Response Thank you for pointing this out. The data has been clarified in the table caption. The data for the Extubation were calculated based on intubated patients and the no. of patients requiring reintubation were based on the no. of extubated patients. (Table 2)

Comment 3(C) Besides, the case numbers of hospital admission in Non-AMS group was 310+119=429, but in the first row: Non-AMS (n=431), the authors should explain the discrepancy). 

Author response: Thank you for pointing this out. Revising our data showed it was a Clerical error. The correct no. is 431 in the Non-AMS group divided into Floor admission (312) and ICU (119) (table 2)

Comment 4:　 In line 211-212 '... patients with AMS were less likely to be African American or have obesity...', but the authors didn't provide possible reasons for the differences. 

B-Besides, I wonder the body surface area might also differed between non-AMS and AMS groups. 

Author response: 

 A- Thank You for raising such an important point, however. Our Data could not explain why the AMS patients were less likely to be African or have obesity, but this strengthens the argument that AMS may be an independent risk factor for poor health outcomes in COVID-19 patients. This point was added as one of the limitations in the study to be addressed by future research. Line (235)

B- We did not have access to the data for body surface area of the patients. So, we cannot address this point. 

Comment 5: According to fig 2, the AMS group was more likely to be intubated and needs mechanical ventilation, but in table 1, the respiratory symptoms of AMS group seemed to be less, especially ' Shortness of breath', this contradiction needs to be discussed. 

Author Response: 

 We agree with this suggestion the contraindication could be explained due to the decreased reporting of presenting complain with AMS Patients according to 

Han JH, Bryce SN, Ely EW, et al. The effect of cognitive impairment on the accuracy of the presenting complaint and discharge instruction comprehension in older emergency department patients. Annals of emergency medicine 2011;57(6):662-71.e2. doi: 10.1016/j.annemergmed.2010.12.002 [published Online First: 2011/01/29]. 

Amended in the discussion (199-201)

---

## [Decision Letter · Decision Letter 1]

6 May 2021

PONE-D-21-02993R1

Altered mental status is a predictor of poor outcomes in COVID-19 patients: A cohort study

PLOS ONE

Dear Dr. Aboueisha,

Thank you for submitting your manuscript to PLOS ONE. After careful consideration, we feel that it has merit but does not fully meet PLOS ONE’s publication criteria as it currently stands. Therefore, we invite you to submit a revised version of the manuscript that addresses the points raised during the review process.

We look forward to receiving your revised manuscript.

Kind regards,

Tai-Heng Chen, M.D.

Academic Editor

PLOS ONE

Reviewers' comments:

Reviewer's Responses to Questions

**Comments to the Author**

1. If the authors have adequately addressed your comments raised in a previous round of review and you feel that this manuscript is now acceptable for publication, you may indicate that here to bypass the “Comments to the Author” section, enter your conflict of interest statement in the “Confidential to Editor” section, and submit your "Accept" recommendation.

Reviewer #1: (No Response)

Reviewer #2: All comments have been addressed

2. Is the manuscript technically sound, and do the data support the conclusions?

Reviewer #1: (No Response)

Reviewer #2: Partly

3. Has the statistical analysis been performed appropriately and rigorously? 

Reviewer #1: (No Response)

Reviewer #2: Yes

4. Have the authors made all data underlying the findings in their manuscript fully available?

Reviewer #1: (No Response)

Reviewer #2: Yes

5. Is the manuscript presented in an intelligible fashion and written in standard English?

Reviewer #1: (No Response)

Reviewer #2: Yes

6. Review Comments to the Author

Reviewer #1: I did not find "Author's response to reviewer recommendation".

My comments were as follows

In this study, authors investigated the association between altered mental status and outcomes in COVID- 19 patients. They found that patients with altered mental status had poor outcomes compared to those patients without altered mental status. Despite the results, I think that there are some concerns.

1. The altered mental status is a vague term and definition. Please clarify the definition of altered mental status.

2. Depth of sedation could be related to altered mental status. Please provide the information of sedative agents.

3. Please provide the causes of altered mental status such as seizure, sepsis associated encephalopathy, metabolic encephalopathy and stroke.

4. When did evaluate the altered mental status during hospitalization? Please provide this information.

Reviewer #2: Comment 1:In the line 81 and 82, 'Patients were divided into two groups: with and

without altered mental status (AMS).' The definition of AMS was not clear. In order to

get a more objective results, an operational definition for AMS is essential.

Author Response Thank you for the comment. The Definition has been clarified in the

manuscript and it is as following: altered mental status (AMS) which encompasses

confusion, amnesia, loss of alertness, disorientation, defects in judgment or thought,

unusual or strange behavior, poor regulation of emotions, and disruptions in

perception, psychomotor skills, and behavior. Line (82-85)

Responses 1: Although the authors described ‘altered mental status (AMS)’ in detail, it’s still difficult to practice in real world. The contents in Line (82-85) were only ‘descriptions’ of AMS, not ‘definitions’. In my opinion, it might be better to use the scores of Glasgow Coma Scale (GCS). In this way, we will be able to quantify AMS. It would also be easier for the audience to realize.

Comment 2:In the line 109 and 110, '...(BMI) was 33.32...with 57.2% being obese and

24.7 being overweight.' It's better to more clearly define 'obese' and 'overweight' . In

addition, obesity is a critical issue for COVID-19 patients, it's also suitable to provide

the BMI of normal population in New Orleans for comparison if possible.

Author Response Thank you for pointing this out. We agree with this comment.

Therefore, we have added the definition of the Obesity and overweight in the results.

(line114,115)

The % of Obese population in Louisiana has been added to the discussion where our

cohort had higher percentage of obese patient than the average of Louisiana

population. Changes has been marked in the manuscript. (215-217)

Responses 2: Agree.

Comment 3(A):The dat of table 2 needs to be carefully corrected. For example, the

meaning of the data in every parentheses made me confusing (e.g. in the first row of

ED disposition: 134 (77) , 31 (72.1); however in the first row of Hospital admission:

310 (72.3), 41(57.7) ; the calculation method of each parentheses was inconsistent !

the inconsistency also noted in the sub-table of Procedures .

Author Response Thank you for pointing this out. The ED depositions data account for patients how came through the Emergency Room a total of 166 in non AMS and 43 for AMS and between the parentheses represent the (%) of the ER cases but these datadon’t show any significance, so it was removed to avoid any confusion. (table 2)

Response 3(A): Agree.

Comment 3(B)the calculation method of each parentheses was inconsistent! the

inconsistency also noted in the sub-table of Procedures

Author Response Thank you for pointing this out. The data has been clarified in the

table caption. The data for the Extubation were calculated based on intubated patients and the no. of patients requiring reintubation were based on the no. of extubated patients. (Table 2)

Response 3(B): Agree.

Comment 3(C)Besides, the case numbers of hospital admission in Non-AMS group

was 310+119=429, but in the first row: Non-AMS (n=431), the authors should explain

the discrepancy).

Author response:Thank you for pointing this out. Revising our data showed it was a

Clerical error. The correct no. is 431 in the Non-AMS group divided into Floor admission (312) and ICU (119) (table 2)

Response 3(C): Floor admission was still 310 in table 2. This should be corrected.

Comment 4: In line 211-212 '... patients with AMS were less likely to be African

American or have obesity...', but the authors didn't provide possible reasons for the

differences.

B-Besides, I wonder the body surface area might also differed between non-AMS and

AMS groups.

Author response:

A-Thank You for raising such an important point, however. Our Data could not explain

why the AMS patients were less likely to be African or have obesity, but this

strengthens the argument that AMS may be an independent risk factor for poor health outcomes in COVID-19 patients. This point was added as one of the limitations in the study to be addressed by future research. Line (235)

Response 4(A): Agree.

B-We did not have access to the data for body surface area of the patients. So, we

cannot address this point.

Responses 4(B): Agree.

Comment 5:According to fig 2, the AMS group was more likely to be intubated and

needs mechanical ventilation, but in table 1, the respiratory symptoms of AMS group

seemed to be less, especially ' Shortness of breath', this contradiction needs to be

discussed.

Author Response:

We agree with this suggestion the contraindication could be explained due to the

decreased reporting of presenting complain with AMS Patients according to

Han JH, Bryce SN, Ely EW, et al. The effect of cognitive impairment on the accuracy of

the presenting complaint and discharge instruction comprehension in older emergency department patients. Annals of emergency medicine 2011;57(6):662-71.e2. doi:10.1016/j.annemergmed.2010.12.002 [published Online First: 2011/01/29].Amended in the discussion (199-201)

Responses 5: Agree.

7. PLOS authors have the option to publish the peer review history of their article (what does this mean?). If published, this will include your full peer review and any attached files.

Reviewer #1: No

Reviewer #2: **Yes: **Chi-Hsiang Chou

---

## [Author Response · Author response to Decision Letter 1]

20 Jun 2021

Response to reviewers' comments

(PONE-D-21-02993)

“Altered mental status is a predictor of poor outcomes in COVID-19 patients: A cohort study.”

Dear Prof. Emily Chenette,

Editor-in-Chief

PLOS ONE Journal 

First, we apologize for delayed response and we are extremely grateful for both reviewers comments and feedback. We would like to thank the editors and reviewer the for giving us the opportunity to submit a revised draft of Our manuscript titled “Altered mental status is a predictor of poor outcomes in COVID-19 patients: A cohort study” to PLOS ONE Journal. 

We appreciate the time and effort that you and the reviewers have dedicated to providing your valuable feedback on our manuscript. We are grateful to the reviewers for their insightful comments on our paper. We have been able to incorporate changes to reflect most of the suggestions provided by the reviewers. We have highlighted the changes within the manuscript. We hope that our responses address the reviewers concerns and wish it meets PLOS journal standards. 

Here is a point-by-point response to the reviewers’ comments and concerns.

Reviewer 1 Comments

Comment 1: The altered mental status is a vague term and definition. Please clarify the definition of altered mental status.

Author Response Thank you for the great input we have tried to the best of our knowledge identify the AMS term as highlighted in the methodology section (Line 83-85). Also, we provided the Mean and SD of both cohort where the AMS group had a GCS of 9.41 ± 4.08 as seen in table 1. 

Comment 2: Depth of sedation could be related to altered mental status. Please provide the information of sedative agents.

Author Response Thank you for the valuable feedback. Sedation will alter the patient level of consciousness, but all our patients had Altered mental status on admission as diagnosed by the attending physician, so they were not on any sedation at the time of diagnosis. This is explained in methodology section in lines (85-86)

Comment 3: Please provide the causes of altered mental status such as seizure, sepsis associated encephalopathy, metabolic encephalopathy and stroke.

Author Response Your feedback is precious. The underlying cause of the AMS is essential to the management plan. In our cohort, we adjusted for neuropsychiatric comorbidities in AMS and non-AMS group, as shown in Figure 2, accounting for the vital role Neuropsychiatric comorbidities could play in developing AMS by performing multivariate analysis patients with COVID-19 who present with AMS have a worse outcome.

The Neuropsychiatric comorbidity could be found in figure 1. 

Comment 4: When did evaluate the altered mental status during hospitalization? Please provide this information.

Author Response Evaluation was performed on Admission. We are grateful for your feedback it has been added to the methodology section in line (85)

Reviewer 2 comments

Comment 1 Although the authors described ‘altered mental status (AMS)’ in detail, it’s still difficult to practice in real world. The contents in Line (82-85) were only ‘descriptions’ of AMS, not ‘definitions. In my opinion, it might be better to use the scores of Glasgow Coma Scale (GCS). In this way, we will be able to quantify AMS. It would also be easier for the audience to realize.

Author response Thank you for your valuable feedback. Adding GCS would be a great addition to the paper; patients enrolled in the study were further assessed using the GCS; the mean and SD of GCS for both groups with and without AMS are present in table 1. we showed that the AMS group had a mean and SD of 9.41 ± 4.08.

Response 3(C): Floor admission was still 310 in table 2. This should be corrected.

Author response It has been corrected 

We are glad that our response answered reviewer 2 previous comments. 

you will find them below

Comment 1:In the line 81 and 82, 'Patients were divided into two groups: with and without altered mental status (AMS).' The definition of AMS was not clear. In order to get a more objective results, an operational definition for AMS is essential. Author Response Thank you for the comment. The Definition has been clarified in the manuscript and it is as following: altered mental status (AMS) which encompasses confusion, amnesia, loss of alertness, disorientation, defects in judgment or thought, unusual or strange behavior, poor regulation of emotions, and disruptions in perception, psychomotor skills, and behavior. Line (82-85)

Responses 1: Although the authors described ‘altered mental status (AMS)’ in detail, it’s still difficult to practice in real world. The contents in Line (82-85) were only ‘descriptions’ of AMS, not ‘definitions’. In my opinion, it might be better to use the scores of Glasgow Coma Scale (GCS). In this way, we will be able to quantify AMS. It would also be easier for the audience to realize.

Answered Above

Comment 2: In the line 109 and 110, '...(BMI) was 33.32...with 57.2% being obese and24.7 being overweight.' It's better to more clearly define 'obese' and 'overweight' . In addition, obesity is a critical issue for COVID-19 patients, it's also suitable to provide the BMI of normal population in New Orleans for comparison if possible.

Author Response Thank you for pointing this out. We agree with this comment.

Therefore, we have added the definition of the Obesity and overweight in the results. (line114,115) The % of Obese population in Louisiana has been added to the discussion where our cohort had higher percentage of obese patient than the average of Louisiana population. Changes has been marked in the manuscript. (215-217)

Responses 2: Agree.

Comment 3(A): The data of table 2 needs to be carefully corrected. For example, the meaning of the data in every parentheses made me confusing (e.g. in the first row of ED disposition: 134 (77) , 31 (72.1); however in the first row of Hospital admission: 310 (72.3), 41(57.7) ; the calculation method of each parentheses was inconsistent ! the inconsistency also noted in the sub-table of Procedures .

Author Response Thank you for pointing this out. The ED depositions data account for patients how came through the Emergency Room a total of 166 in non AMS and 43 for AMS and between the parentheses represent the (%) of the ER cases but these data don’t show any significance, so it was removed to avoid any confusion. (table 2)

Response 3(A): Agree.

Comment 3(B)the calculation method of each parentheses was inconsistent! the

inconsistency also noted in the sub-table of Procedures

Author Response Thank you for pointing this out. The data has been clarified in the table caption. The data for the Extubation were calculated based on intubated patients and the no. of patients requiring reintubation were based on the no. of extubated patients. (Table 2)

Response 3(B): Agree.

Comment 3(C)Besides, the case numbers of hospital admission in Non-AMS group was 310+119=429, but in the first row: Non-AMS (n=431), the authors should explain the discrepancy).

Author response: Thank you for pointing this out. Revising our data showed it was a Clerical error. The correct no. is 431 in the Non-AMS group divided into Floor admission (312) and ICU (119) (table 2)

Response 3(C): Floor admission was still 310 in table 2. This should be corrected.

Answered above 

Comment 4: In line 211-212 '... patients with AMS were less likely to be African

American or have obesity...', but the authors didn't provide possible reasons for the differences.

B-Besides, I wonder the body surface area might also differed between non-AMS and AMS groups.

Author response:

A-Thank You for raising such an important point, however. Our Data could not explain why the AMS patients were less likely to be African or have obesity, but this strengthens the argument that AMS may be an independent risk factor for poor health outcomes in COVID-19 patients. This point was added as one of the limitations in the study to be addressed by future research. Line (235)

Response 4(A): Agree.

B-We did not have access to the data for body surface area of the patients. So, we

cannot address this point.

Responses 4(B): Agree.

Comment 5:According to fig 2, the AMS group was more likely to be intubated and needs mechanical ventilation, but in table 1, the respiratory symptoms of AMS group seemed to be less, especially ' Shortness of breath', this contradiction needs to be discussed.

Author Response:

We agree with this suggestion the contraindication could be explained due to the

decreased reporting of presenting complain with AMS Patients according to

Han JH, Bryce SN, Ely EW, et al. The effect of cognitive impairment on the accuracy of the presenting complaint and discharge instruction comprehension in older emergency department patients. Annals of emergency medicine 2011;57(6):662-71.e2. doi:10.1016/j.annemergmed.2010.12.002 [published Online First: 2011/01/29].Amended in the discussion (199-201)

Responses 5: Agree.

---

## [Decision Letter · Decision Letter 2]

22 Jul 2021

PONE-D-21-02993R2

Altered mental status is a predictor of poor outcomes in COVID-19 patients: A cohort study

PLOS ONE

Dear Dr. Aboueisha,

Thank you for submitting your manuscript to PLOS ONE. After careful consideration, we feel that it has merit but does not fully meet PLOS ONE’s publication criteria as it currently stands. Therefore, we invite you to submit a revised version of the manuscript that addresses the points raised during the review process.

We look forward to receiving your revised manuscript.

Kind regards,

Tai-Heng Chen, M.D.

Academic Editor

PLOS ONE

Journal Requirements:

Reviewers' comments:

Reviewer's Responses to Questions

**Comments to the Author**

1. If the authors have adequately addressed your comments raised in a previous round of review and you feel that this manuscript is now acceptable for publication, you may indicate that here to bypass the “Comments to the Author” section, enter your conflict of interest statement in the “Confidential to Editor” section, and submit your "Accept" recommendation.

Reviewer #1: All comments have been addressed

Reviewer #2: All comments have been addressed

2. Is the manuscript technically sound, and do the data support the conclusions?

Reviewer #1: Yes

Reviewer #2: Partly

3. Has the statistical analysis been performed appropriately and rigorously? 

Reviewer #1: Yes

Reviewer #2: Yes

4. Have the authors made all data underlying the findings in their manuscript fully available?

Reviewer #1: Yes

Reviewer #2: Yes

5. Is the manuscript presented in an intelligible fashion and written in standard English?

Reviewer #1: Yes

Reviewer #2: Yes

6. Review Comments to the Author

Reviewer #1: Thank you for your revised manuscript.

Authors modified and updated the manuscript based on the reviewer's comments.

Reviewer #2: I suggested the authors to provide the Glasgow Coma Scale (GCS) score of the AMS patients previously. However, I still cannot get any GCS data by this revised article. I think this point needs to be well explained.

7. PLOS authors have the option to publish the peer review history of their article (what does this mean?). If published, this will include your full peer review and any attached files.

Reviewer #1: No

Reviewer #2: No

---

## [Author Response · Author response to Decision Letter 2]

8 Sep 2021

Response to reviewers' comments

(PONE-D-21-02993)

“Altered mental status is a predictor of poor outcomes in COVID-19 patients: A cohort study.”

Dear Prof. Emily Chenette,

Editor-in-Chief

PLOS ONE Journal 

First, we apologize for delayed response because of hurricane Ida and the power loss afterward. we are extremely grateful for both reviewers’ comments and feedback. We would like to thank the editors and reviewer for giving us the opportunity to submit a revised draft of Our manuscript titled “Altered mental status is a predictor of poor outcomes in COVID-19 patients: A cohort study” to PLOS ONE Journal. 

We appreciate the time and effort that you and the reviewers have dedicated to providing your valuable feedback on our manuscript. We are grateful to the reviewers for their insightful comments on our paper. We have been able to incorporate changes to reflect most of the suggestions provided by the reviewers. We have highlighted the changes within the manuscript. We hope that our responses address the reviewers concerns and wish it meets PLOS journal standards. 

Here is a point-by-point response to the reviewers’ comments and concerns.

Reviewer 2 Comments

Comment I suggested the authors to provide the Glasgow Coma Scale (GCS) score of the AMS patients previously. However, I still cannot get any GCS data by this revised article. I think this point needs to be well explained.

Answer Thank you for your comments and reviews, we have edited the methodology section to show how used GCS and edited the results to highlight this finding 

Line 91-92

Line 129-130

---

## [Decision Letter · Decision Letter 3]

20 Sep 2021

Altered mental status is a predictor of poor outcomes in COVID-19 patients: A cohort study

PONE-D-21-02993R3

Dear Dr. Aboueisha,

We’re pleased to inform you that your manuscript has been judged scientifically suitable for publication and will be formally accepted for publication once it meets all outstanding technical requirements.

Kind regards,

Tai-Heng Chen, M.D.

Academic Editor

PLOS ONE

Reviewers' comments:

Reviewer's Responses to Questions

**Comments to the Author**

1. If the authors have adequately addressed your comments raised in a previous round of review and you feel that this manuscript is now acceptable for publication, you may indicate that here to bypass the “Comments to the Author” section, enter your conflict of interest statement in the “Confidential to Editor” section, and submit your "Accept" recommendation.

Reviewer #2: All comments have been addressed

2. Is the manuscript technically sound, and do the data support the conclusions?

Reviewer #2: Yes

3. Has the statistical analysis been performed appropriately and rigorously? 

Reviewer #2: Yes

4. Have the authors made all data underlying the findings in their manuscript fully available?

Reviewer #2: Yes

5. Is the manuscript presented in an intelligible fashion and written in standard English?

Reviewer #2: Yes

6. Review Comments to the Author

Reviewer #2: The description "(9.41 ± 4.0815.00 ± 0.00, p<0.001)" on line 129-130 should be corrected. (I think the correct description might be "(9.41 ± 4.08 versus 15.00 ± 0.00, p<0.001)".

7. PLOS authors have the option to publish the peer review history of their article (what does this mean?). If published, this will include your full peer review and any attached files.

Reviewer #2: No

---

## [Editor Report · Acceptance letter]

27 Sep 2021

PONE-D-21-02993R3 

Altered mental status is a predictor of poor outcomes in COVID-19 patients: A cohort study. 

Dear Dr. Aboueisha:

I'm pleased to inform you that your manuscript has been deemed suitable for publication in PLOS ONE. Congratulations! Your manuscript is now with our production department. 

Kind regards, 

on behalf of

Dr. Tai-Heng Chen 

Academic Editor

PLOS ONE